# Reproducibility study - Counterfactual Generative Networks

¹ ## Reproducibility Summary

² **Scope of Reproducibility** In this study, we worked on the reproducibility of the results in the paper Counterfactual
³ Generative Networks by Axel Sauer, Andreas Geiger. The study is performed based on the following claims;

⁴ • Counterfactual generative network (CGN) can generate high-quality counterfactual images with direct control
⁵ over shape, texture, and background.

⁶ • Using generated counterfactual images in training data set improves the classifier's out-of-domain robustness.

⁷ • Using generated counterfactual images in the training data set only marginally degrades overall accuracy.

⁸ **Methodology** Source code used in the original paper was already provided by the authors and implemented in Pytorch.
⁹ Code was adapted for different experimentation purposes. Additionally, authors used some pre-trained networks in their
¹⁰ experiments. Original paper includes a link to these networks' implementation as well.

¹¹ **Results** We managed to reproduce most of the results in the original paper. We had some difficulties reproducing the
¹² first claim, but the results of our experiments support the second and the third claim.

¹³ **What was easy** The architecture of the networks was explained clearly in the paper and it was relatively easy to
¹⁴ comprehend. Implementation-wise, the code was clean enough to run without requiring an extensive debugging.
¹⁵ Appendix in the paper provided quite many visualization and detailed explanation regarding the experiments, including
¹⁶ the failed cases. This gave us an insight about the limitations in the models' performance.

¹⁷ **What was difficult** Main difficulty in the experiments was that the computation time required for the model training with
¹⁸ ImageNet data set. It is approximated to take about 214 hours to conduct a single experiment, while running on a cluster
¹⁹ computing system. To complete the experiments in the given time frame, subset of the ImageNet (ImageNet-mini) is
²⁰ used.

²¹ **Communication with original authors** We contacted with the authors regarding the discrepancies between the code
²² and the paper, and the unaligned results. Authors clarified our concerns several times in different occasions.

# 1   Introduction

Deep neural networks (DNNs) are the fundamental learning algorithms which are widely used in the field of machine learning. Although the DNNs perform well in many tasks, they still struggle with handling the unseen data. Data bias is seen as the main factor in the failed cases. If the model has always seen a particular object in a certain background, it tends to correlate the object with the background, and the model fails to recognize the object when it appears in a different background, which is a significant obstacle towards having a generalized model.

Data augmentation is seen as a strong regularizer and an efficient way to extend the training data set in machine learning algorithms, by Wong et al. (2016). Previous studies by Goyal et al. (2018) showed that data augmentation with synthetic images is a promising solution. Authors designed a new generative model called counterfactual generative network (CGN), which generates counterfactual images based on two main assumptions; independent mechanism and the composition mechanism. Independent mechanism suggests that the modules that generate the synthetic image are independent of each other (dos Santos Tanaka and Aranha (2019)). This way spurious correlation can be minimized.

CGN generates an image by combining three independent components which are defined as shape, texture and background, and those components are combined analytically, following a certain equation. In this paper, same assumptions are accepted for the sake of assessing the reproducibility of the experiment results.

# 2   Scope of reproducibility

Focus of our work is to reproduce the general trends in the experimental results, such as an increase in the performance of the classifier which is trained on counterfactual images in addition to the original data set. In this study, we worked to validate the following claims in the original paper;

1. CGN can generate high-quality counterfactual images with direct control over shape, texture, and background.
2. Using generated counterfactuals in the training data set improves the classifier's out-of-domain robustness
3. Using generated counterfactuals in training data set only marginally degrades overall accuracy

Claim 1, which is supported by experiments found in Section 3.4.1 and Fig 4, 2, 9, 8 was not proven correct. Second claim, which is assisted by experiments described in Section 3.4.2 and Table 2 was found correct. Finally, third claim is supported by Section 3.4.3 and Table 1 and 5 and 4 and proven correct. In the following sections, methodology we adopted in this study is explained. Following that, claims and the results are presented. Finally, we discus the strengths and the weaknesses of the original paper in the discussion section.

# 3   Methodology

## 3.1   Implementation

The base code [1] is provided by the authors in the original paper. It was well documented and sufficiently clear to run without requiring any debugging.

## 3.2   Model descriptions

To investigate claims made by the authors in the original papers, we carried out experiments using counterfactual generative networks (CGNs). In this section we describe the architectures of the two different CGNs we used.

**MNIST CGN**   : In the original paper, it is assumed that the generative process of the counterfactual images can be decomposed into three independent mechanisms; shape, texture and background mechanism. For the MNIST data set, mechanisms for texture and background are designed with the exact same structure, while shape mechanism has a slightly different structure.

---

[1]`https://github.com/autonomousvision/counterfactual_generative_networks`
[2]This image is taken from the original paper Sauer and Geiger (2021).

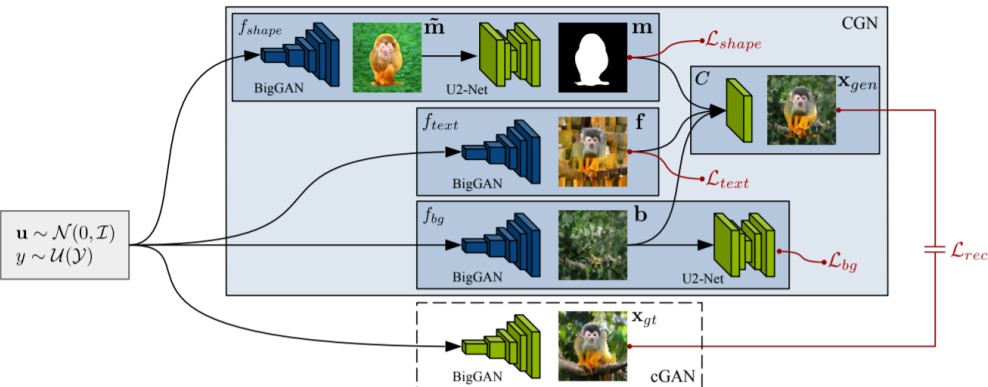

Figure 1: **Counterfactual Generative Network (CGN)** Here, the architecture of the CGN is displayed. The network consists of four main mechanisms. These are $f_{shape}$ for shape component, $f_{text}$ for texture, $f_{bg}$ for background and $\mathcal{C}$ for composition. Pretrained models are shown in green, while the models with trainable parameters are shown in blue. Performance of the CGN is assessed via the reconstruction loss, which is computed using the conditional GAN (cGAN) outputs. In the model, cGAN took part only in the training process. Each mechanism receives a Gaussian noise vector $\mathbf{u}$ and a label $\mathcal{Y}$. The loss values, which is shown in red, are minimized during the training process. Counterfactual images are generated using a noise vector and independently sampled labels (one label for each mechanism)[2].

In the generation of a new image, all three components are merged based on a certain composition mechanism, which is the second assumption in the original paper. Most important feature of the composition mechanism is that it is defined analytically, not learned by any model. Composition mechanism is defined as follows.

$$x_{gen} = C(\mathbf{m}, \mathbf{f}, \mathbf{b}) = \mathbf{m} \odot \mathbf{f} + (1 - \mathbf{m}) \odot \mathbf{b} \tag{1}$$

where the m is the mask (shape component), f is the foreground and b is the background. $\odot$ is used to represent the element-wise multiplication.

**ImageNet CGN**  Architecture of the CGN designed for ImageNet is displayed in Fig 1. Independent mechanism and composition mechanism assumptions are applied in the ImageNet CGN as well. Additionally, several loss values are computed to take part in the model training. Since the mechanisms that comprise the generative model are independent of each other, each mechanism has its own loss value. Loss values can be found in red text in Figure 1. $\mathcal{L}_{shape}$, $\mathcal{L}_{text}$, $\mathcal{L}_{bg}$ correspond to the loss in shape, texture, background mechanisms, respectively, and $\mathcal{L}_{rec}$ corresponds to the reconstruction loss which is computed using the output of the generative model and the output of the pre-trained BigGAN model. In the training process, all loss values are linearly combined and jointly optimized. Overall loss is calculated as follows.

$$\begin{aligned} \mathcal{L} &= \mathcal{L}_{rec} + \mathcal{L}_{shape} + \lambda_5 \mathcal{L}_{text} + \lambda_6 \mathcal{L}_{bg} \\ &= \lambda_1 \mathcal{L}_{L1} + \lambda_2 \mathcal{L}_{perc} + \lambda_3 \mathcal{L}_{binary} + \lambda_4 \mathcal{L}_{mask} + \lambda_5 \mathcal{L}_{text} + \lambda_6 \mathcal{L}_{bg}^{\phantom{x}3} \end{aligned} \tag{2}$$

where $\lambda_1 = 100, \lambda_2 = 5, \lambda_3 = 300, \lambda_4 = 500, \lambda_5 = 5, \lambda_6 = 2000$. Authors did not provide any information regarding the calculation of the $\lambda$ values.

Different than the MNIST dataset, trainable BigGAN models are placed in each independent mechanism. Furthermore, shape mechanism and background mechanism comprise a pre-trained U2-Net to process the output of the BigGAN models. Additionally, another pre-trained BigGAN model is used to generate non-counterfactual images using the given noise and the label, and it is mainly used to compute the reconstruction error and train the CGN.

### 3.3   Datasets

Two main datasets are used in the experiments; MNIST and ImageNet-1k. MNIST data set consists of three subsets; colored-MNIST, double-colored MNIST and the wildlife MNIST. Description of the datasets can be found below.

---

[3]This equation is provided by the original paper Sauer and Geiger (2021).

**Colored MNIST:** It has images 50k for training and 6k images for testing set. Ten different colors are selected and each color is assigned to a single type of digit as their mean color in the training set. In the test set, the colors are randomly assigned to the digits.

**Double-colored MNIST:** It has 50k for samples for training and 6k samples for testing. The main difference with the colored MNIST is that the background is also colored with the one of the selected colors.

**Wildlife MNIST:** It has 50k for training and 6k for testing. Ten distinct texture images are selected from the striped class which is provided by Cimpoi et al. (2013) for the texture. Ten other distinct texture images are selected from the veiny class from the same source, for the background.

**ImageNet-mini:** It has 1k classes, 34.752 samples for training and 10k samples for validation.

### 3.4 Experimental setup

In this section we describe the experiments performed to investigate the claims described in section 2. For comprehensibility, we list the claims and corresponding experiments. In this study, MNIST models are trained on a single RTX 1080 GPU located in a cluster computer system. A device with more memory, such as an Titan RTX, is advised when training ImageNet models.

#### 3.4.1 Claim 1

The following are experiments carried out to investigate the claim *'A CGN can generate high-quality counterfactual images with direct control over shape, texture, and background.'*

**MNIST counterfactuals** We trained 3 CGN's, each on one of the MNIST datasets (described in Section **??**) using code provided by the authors, which can be found and run HERE. The architecture of the CGN's trained on the MNIST datasets is described in Section 3.2. During training we sample counterfactuals generated by the model and we compare counterfactuals generated by the trained CGN to examples in Sauer and Geiger (2021). Training the CGN was done with the default parameters that the authors also used in the paper.

**ImageNet counterfactuals** To produce ImageNet counterfactuals from a class conditional variable and random vector we regressed a CGN with the pre-trained BigGAN backbone, as defined in Section 3.2. We investigate the quality of the generative network in two ways: first by visual inspection and, secondly, measure the Inception Score and mask mean $\mu_{mask}$. Further, during training, we closely monitor the elements of the compositions to confirm the intended loss behaviour. The hyperparameters for the losses and learning rates are provided by the authors. This includes the lambdas defined in Equation 2 and learning rates 8E-6, 3E-5 and 1E-5 for shape, texture and background, respectively. For our single GPU with 24GB memory the highest possible batch size was 5, which requires changing the number of episodes and batch accumulation to 200 and 500, respectively. To this end, we regress $5 \cdot 500 \cdot 200 = 5 \cdot 10^5$ unique images taking approximately 30 hours. Finally, we experiment with a modified texture mechanism where the patch grid is created by filling the image randomly with the object till a degree is met.

**Inception Score and $\mu_{\mathbf{mask}}$** To assess the quality of the generated images from the ImageNet model, we calculate the Inception Score[4] as introduced by Salimans et al. (2016) for a uniform sample of non-counterfactual images. The authors didn't state in their paper how many samples they used. So, we chose to use 50,000 images with no splits as Barratt and Sharma (2018) suggest that this is an appropriate amount of images given the number of classes in ImageNet. The generated images from the CGN include a mask for each image of which we compute the mean pixel value $\mu_{mask}$.

We calculated the inception score for our self trained CGN, the CGN using the weights provided by the authors and a pretrained BigGAN model. As BigGAN does not generate masks, the $\mu_{mask}$ value was determined only for the pretrained and self trained CGNs.

#### 3.4.2 Claim 2

The following are experiments carried out to investigate the claim *'Including generated counterfactuals in the training data set improves the classifier's out-of-domain robustness.'*

---

[4]The following TensorFlow implementation of the Inception Score was used: https://github.com/tsc2017/Inception-Score

**Classifying MNIST datasets**    Like the authors of the original paper, we trained a classifier on MNIST data and compared the performance of the classifier for different compositions of the training data. We compare classifiers trained on original datasets, original datasets with counterfactuals produced by the trained CGN, original dataset with non-counterfactual samples generated by the CGN and only on non-counterfactual samples generated by the CGN. We test on counterfactuals generated manually, so not by any CGN. We also test on non-counterfactual samples. We used a classifier that has the same architecture as the one used by the authors, with the default parameters in the code provided by the authors. We test how the testing accuracy changes with different datasizes and differing ratios of number of counterfactuals in the training data. As an additional experiment, we produced visual explanations for the classifiers trained on double coloured MNIST. These visual explanations were produced using GradCAM by Selvaraju et al. (2016) applied onto the last convolutional layer of the model.

**ImageNet-mini classification**    Similar to the authors, we measure the out-of-domain robustness for the ResNet-50 classifier with the ImageNet-9 background challenge dataset Xiao et al. (2020). This dataset contains two subsets that hold images with randomized backgrounds of the same class and of different classes. Dubbed mixed-same and mixed-rand, their difference in classification top-1 accuracy is a solid measure of class background dependence. Not similar to the authors, we performed classification training on a subset of ImageNet named ImageNet-mini by Figotin (2020). This dataset contains fewer images per class, significantly decreasing convergence time while only marginally dropping accuracy. With the ImageNet-mini and ImageNet-9 dataset we train the ResNet-50 classifier in three ways. Setting (1) contains ImageNet-mini training data only, (2) adds counterfactuals produced with the authors CGN weights, and (3) adds counterfactuals produced by our weights. The amount of random counterfactuals produced for training was $10^5$ and is suspected to be sufficient. The hyperparameters to train the classifier were searched to obtain high accuracies on ImageNet-mini and are presented in 7. The search concludes on learning rate 1E-4 and counterfactual ratio 2.0, where momentum is 0.9 and weight decay is 1E-4. Convergence point for each setting is also presented for ease of verification and were all reached within 5 hours on batch size 32.

### 3.4.3   Claim 3

The following are experiments carried out to investigate the claim *'Including generated counterfactuals in training data set only marginally degrades overall accuracy'*

**Classifying MNIST datasets**    We also tested the performance of classifiers trained on the original dataset to performance of the classifier trained on counterfactuals (with or without original data) with non-counterfactual samples as test data. This tells us whether training a classifier with counterfactuals affects performance on in-domain (non-counterfactual) data.

**ImageNet-mini classification**    Copying the experimental setting for classification training in claim 2, we test the classifier on the base unmodified images of ImageNet-9 for in-domain accuracies. Additionally we inspect the top-1 and top-5 accuracies of our ensemble classifier on ImageNet-mini itself.

Besides in-domain we investigate the shape-biases of the resulting classifier ensemble with the Cue Conflict dataset Gatys et al. (2015). The dataset consists of images that are generated by mixing a random texture and random object in an iterative style manner. Higher bias indicates classification based on object, whereas a lower bias indicates classification on texture. This helps us answer whether we can control the separate heads for shape, texture and background while maintaining in-domain accuracy.

Since this experiment is evaluated on the same settings as claim 2, the evaluation was performed simultaneously. Not increasing computation time.

## 4   Results

In this section we describe the results of the experiments described in the methodology. We find that our experiments support claim 2 and claim 3, but we didn't find sufficient support for claim 1 across the datasets.

### 4.1 Claim 1

In the following sections we describe results of experiments carried out to investigate the claim *'A CGN can generate high-quality counterfactual images with direct control over shape, texture, and background.'*.

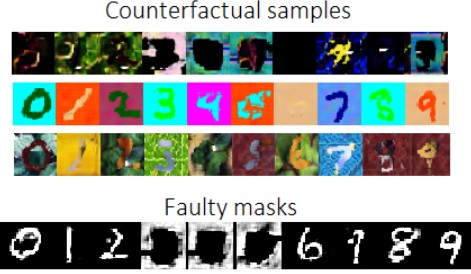

Figure 2: Samples generated by the CGN trained on the three MNIST datasets. For each of the three dataset, counterfactuals are shown. Especially colored MNIST and wildlife are of low quality. The last row shows faulty masks that were learned by the CGN.

**MNIST counterfactuals**   By running the code for training the CGN several times, we eventually managed to train a CGN that produced satisfactory samples both for the MNIST wildlife and MNIST double colored dataset. However, after attempting to train the CGN at least 10 times with different initialisation weights, the results were not satisfactory for the colored MNIST dataset. Some examples are shown in Figure 2. As shown in the Figure, the CGN learns faulty masks for some of the digits for the colored MNIST dataset. When asked whether the authors experienced similar problems training the MNIST CGN, the response was that they experienced the same, but sometimes managed to train a CGN with satisfactory results. Figure 2 shows some of the faulty masks that were learned by the CGN. The results for the double-colored and wildlife samples seem satisfactory, although the masks learned for the digits four and five are not of high quality. The authors do not specify what percentage of trials yielded satisfactory results. We found that approximately 2/3 of training trials yielded results that are satisfactory for double colored MNIST and wildlife MNIST. No good training trials were found for colored MNIST. In Appendix A, more examples can be found.

**ImageNet counterfactuals**   The quality of the compositional network is presented in Table 3 by the Inception Score and mean mask value. Training the CGN ourselves obtains an IS of $115.5$ compared to their trained weights $129.4$, indicating rather low generative capabilities relative to the BigGAN. In Appendix F we investigate the three CGN mechanisms and three problems they can cause. Our IS results align well with that of the original paper $130.2$. The reported $\mu_{\mathbf{mask}}$ of $0.33$ indicates no mask collapse and is within the deviation shown by the authors $0.3 \pm 0.2\%$. Although quality of counterfactuals are correlated with the non-counterfactual IS, we additionally investigate 16 randomized counterfactual images shown in Figure 7 of Appendix E. Visually the generated counterfactuals appear unreal, but the quality we are after is an accurate shape, texture and background on the conjoined image. These random counterfactuals appear to show these qualities.

### 4.2 Claim 2

In the following sections we describe results of experiments carried out to investigate the claim *'Including generated counterfactuals in the training data set improves the classifier's out-of-domain robustness.'*

**Classifying MNIST datasets**   In Table 2, we show the results of training classifiers on several different datasets. When trained on only the original dataset, performance on the testset is $39\%$ for colored MNIST, but only around $10\%$ for both double-colored and wildlife. This was expected and similar to what the authors found. When trained on only counterfactual data or a combination of both original data and counterfactual data, performance increases significantly. While performance increases compared to performance when trained only on original data, it doesn't reach performance reported in the original paper, except for double-colored MNIST. However, performance is proportional to the quality of our CGN, since especially colored MNIST and wildlife MNIST were difficult to train the CGN on (see also the results in Figure 2).

**ImageNet-mini classification** The accuracies for ImageNet-9 background challenge are presented in Table 5. Our obtained out-of-domain robustness is highest when training on ImageNet-mini and author provided CGN weights, with a background gap of 7.7%. This cannot be compared with the authors presented value of 3.3% since a different dataset is used. Instead, we compare the difference in accuracy when including counterfactuals. This difference is $-1.2\%$, indicating an increased out-of-domain robustness, equal to the authors reported $-1.2\%$. Using our own weights produces lower accuracies, but the difference seems negligible.

**Gradient heatmaps** Figure 3 shows that when the classifier is trained on the non-counterfactual double colored dataset the positive gradients in the last convolutional layer are correlated with the color theme used in the image. However, when trained on the double colored counterfactual dataset the gradients only slightly vary when changing the color theme. We also observe that a significant part of the heatmaps from the double colored original classifier are empty, indicating that the gradient is not positive for the whole layer.

### 4.3 Claim 3

In the following sections we describe results of experiments carried out to investigate the claim *'Including generated counterfactuals in training data set only marginally degrades overall accuracy'*

**Classifying MNIST datasets** In Table 1 we show that training the classifier on counterfactuals as well as on original data only marginally decreases the accuracy on the test data when the test data consists of in-domain samples.

**ImageNet-mini classification** Since claim 3 is evaluated on the same training configuration of claim 2, we present the in-domain accuracies of unmodified ImageNet-9 images in Table 5. There appears no drop in accuracy when using authors weights $0.0\%$, which does not align with their reported drop of $1.4\%$ on IN-9. Using are our own weights we achieve comparable results on IN-9.

Table 4 shows the shape bias and classifier ensemble accuracies on ImageNet-mini. When the classifier is trained with counterfactuals the texture head is able achieve a higher top-1 accuracy of $+3.4\%$ and shows an appropriately low shape bias. This shows we can individually control the three heads for classification. However, our results are not comparable with the authors since a different dataset is used. But results of shape bias are similar for our dataset.

| | Colored MNIST | | Double colored MNIST | | Wildlife MNIST | |
|---|---|---|---|---|---|---|
| | Test accuracy | Train accuracy | Test accuracy | Train accuracy | Test accuracy | Train accuracy |
| Original | 99.8 | 99.8 | 100.0 | 97.6 | 100.0 | 99.9 |
| Original + CGN | 99.8 | 94.7 | 99.9 | 94.4 | 99.6 | 99.1 |

Table 1: Accuracies for classifiers trained original or original and counterfactual data, and tested on test data containing in-domain images.

| | Colored MNIST | | Double colored MNIST | | Wildlife MNIST | |
|---|---|---|---|---|---|---|
| | Test accuracy | Train accuracy | Test accuracy | Train accuracy | Test accuracy | Train accuracy |
| Original | 39.0 | 99.8 | 10.1 | 100.0 | 10.6 | 99.4 |
| GAN | 11.7 | 95.4 | 10.0 | 98.7 | 10.8 | 95.1 |
| Original + GAN | 38.1 | 99.9 | 10.1 | 100.0 | 10.7 | 100.0 |
| CGN | 32.5 | 90.0 | 87.3 | 92.8 | **70.2** | 99.1 |
| Original + CGN | **53.5** | 91.1 | **87.9** | 94.4 | 63.2 | 97.0 |

Table 2: Test and training accuracy of classifiers trained on several different datasets. The testset consisted of counterfactual data. The counterfactuals used in the training data were generated by a CGN that was trained by us.

| Model | IS | $\mu_{mask}$ |
|---|---|---|
| CGN (theirs) | 129.4 | 0.332 |
| CGN (ours) | 115.5 | 0.286 |
| BigGAN | 195.9 | - |

Table 3: Inception Score and mean mask $\mu_{\textbf{mask}}$ of CGN.

| Trained on | Shape Bias | top-1 IN Acc | top-5 IN Acc |
|---|---|---|---|
| ImageNet-mini | 29.1 % | 65.7 % | 88.2 % |
| IN-mini + CGN/Shape | 49.6 % | | |
| IN-mini + CGN/Text | 18.0 % | 69.1 % | 88.1 % |
| IN-mini + CGN/Bg | 23.1 % | | |

Table 4: Shape bias of the shape, texture and background classification heads. With accuracies on ImageNet-Mini.

| | top-1 Test Accuracies | | | |
|---|---|---|---|---|
| Trained on | IN-9 | Mixed-Same | Mixed-Rand | BG-Gap |
| ImageNet (base) | 94.7 % | 85.6 % | 78.1 % | 7.5 % |
| IN-mini | **91.6** % | 81.8% | 73.3 % | 8.5 % |
| IN-mini + CGN | **91.6** % | **81.9** % | **74.2** % | **7.7** % |
| IN-mini + our CGN | 89.7 % | 81.3 % | 72.7 % | 8.6 % |

Table 5: ImageNet-9 accuracy with and without counterfactuals.

Figure 3: GradCAM heatmaps for our classifier trained on the double colored MNIST dataset. The x-position in the grid determines the color theme used for the image and the y-position determines the shape. The numbers above the x-axis correspond to the color theme used for that specific digit in the in-domain dataset.

## 5 Discussion

**Claim 1** The first claim, *CGNs can generate high-quality counterfactual images with direct control over shape, texture, and background*, was not trivial to reproduce. In particular, we found that is was difficult to train CGNs on the MNIST dataset that can generate satisfactory images. The authors of the original paper likely kept trying for longer. However, the difficulty we had with training the CGN might indicate that the architecture of the CGN can be improved to make the training process easier and get better results.

Our experiments with ImageNet yielded very similar and stable results. Although our quality of the generated images is slightly lower than that of the authors of the original paper. We remain within an acceptable lower inception score. The reason for this might be the training time. However, when increasing training time, failure cases such as background residue become more common, this is further discussed in Appendix F.

**Claim 2** The claim *Including generated counterfactuals in the training data set improves the classifier's out-of-domain robustness* is supported by our experiments. For the MNIST datasets, Table **??** shows that adding counterfactual data to the training data improves accuracy on out-of-domain test data.

Figure 3 further supports claim 2, as we observe that the positive gradients in the final layer for the classifier trained on counterfactuals do not spuriously correlate to the color theme used in the image, in contrast to the classifier trained on the original dataset.

For ImageNet-mini, the decreased BG-gap shows that when adding counterfactual data to training data, the classifier's out-of-domain robustness is increased. Although using our own weights does lead to lower scores across the test-set, which was expected due to the lower IS score of the CGN.

**Claim 3** The claim *'Including generated counterfactuals in training data set only marginally degrades overall accuracy'* is supported by our experiments.

Our MNIST experiments show that adding counterfactuals barely decreases accuracy on in-domain data. The accuracies we find are lower than those of the authors. This is likely due to the fact that the quality of the CGNs that we were able to train is lower than that of the authors.

For ImageNet, we show no degradation in classifier accuracy on the unmodified IN-9 test set. This is not similar to the authors and is most likely caused by our choice of smaller sized ImageNet-mini subset. The CGN artificially increases dataset size and therefore mainly helps on smaller datasets. Further we have shown similar biases across the classifier ensemble indicating full control off the classifier decision making. With the texture classification head we report an increase in top-1 accuracy on in-domain ImageNet-mini, but is also most likely caused by the artificial increase in dataset size.

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

# 6 Appendix

278 ## A MNIST counterfactuals generated by our CGNs

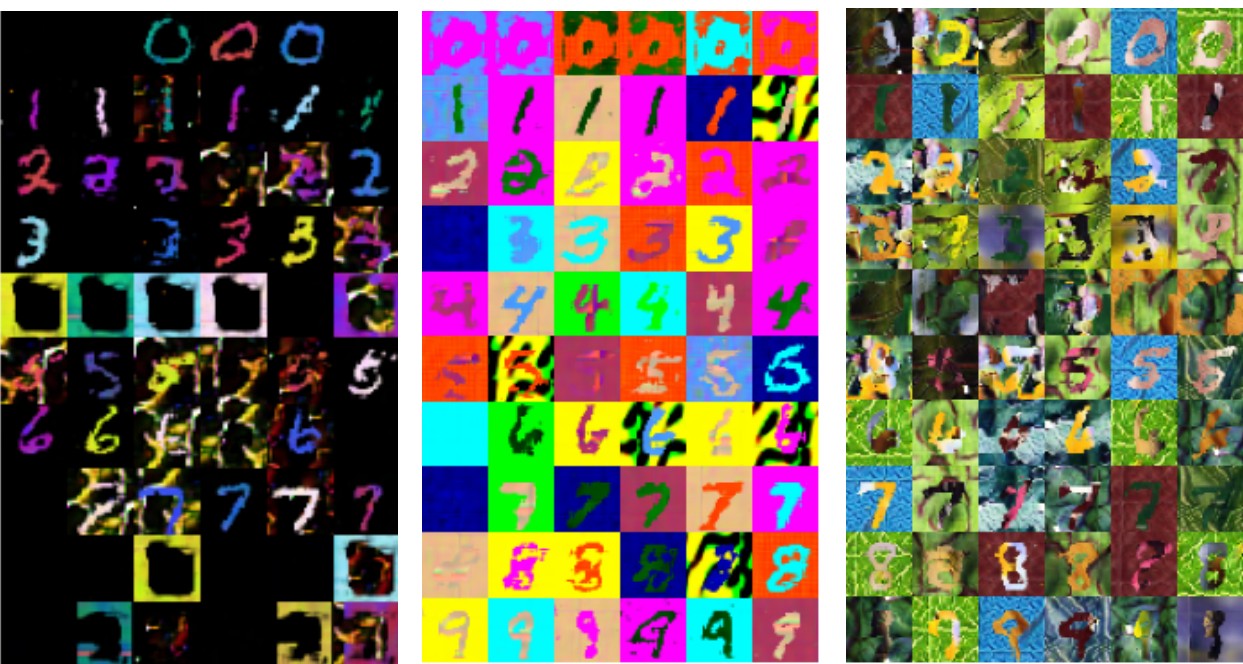

Figure 4: **MNIST counterfactuals.** From left to right: colored, double-colored and wildlife MNIST. Images are generated by the CGNs which were trained by us.

279 ## B MNIST classification with weights of authors original paper

| | colored MNIST | | double-colored MNIST | | wildlife MNIST | |
|---|---|---|---|---|---|---|
| | Train Acc | Test Acc | Train Acc | Test Acc | Train Acc | Test Acc |
| Original | 99.7 % | 38.9 % | 100.0 % | 10.1 % | 99.3 % | 10.6 % |
| GAN | 99.9 % | 25.83 % | 100.0 % | 9.9 % | 100.0 % | 10.8 % |
| Original + GAN | 99.8 % | 40.3 % | 100.0 % | 10.0 % | 100.0 % | 10.7 % |
| CGN | 99.3 % | 92.8 % | 96.7 % | 90.2 % | 98.5 % | 84.4 % |
| Original + CGN | 99.2 % | 96.4 % | 97.2 % | 87.2 % | 98.0 % | 76.24 % |

Table 6: Test and training accuracy of classifiers trained on several different datasets. The testset consisted of counterfactual data. The counterfactuals used in the trainingdata were generated by a CGN with weights provided by the authors of the original paper.

 # C   MNIST ablation study

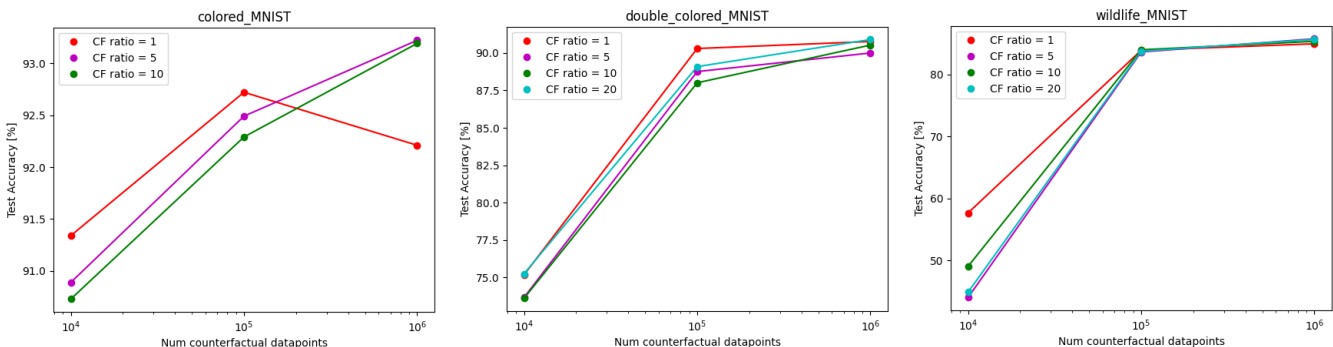

Figure 5: Test accuracy for classifiers trained on original data and counterfactual data with different CF ratios. The CF ratio indicates how many counterfactuals we generate per sampled noise. For colored MNIST, the maximum CF ratio is ten as there are only ten possible colors per shape. The counterfactuals in the trainingdata were generated by a CGN we trained ourselves.

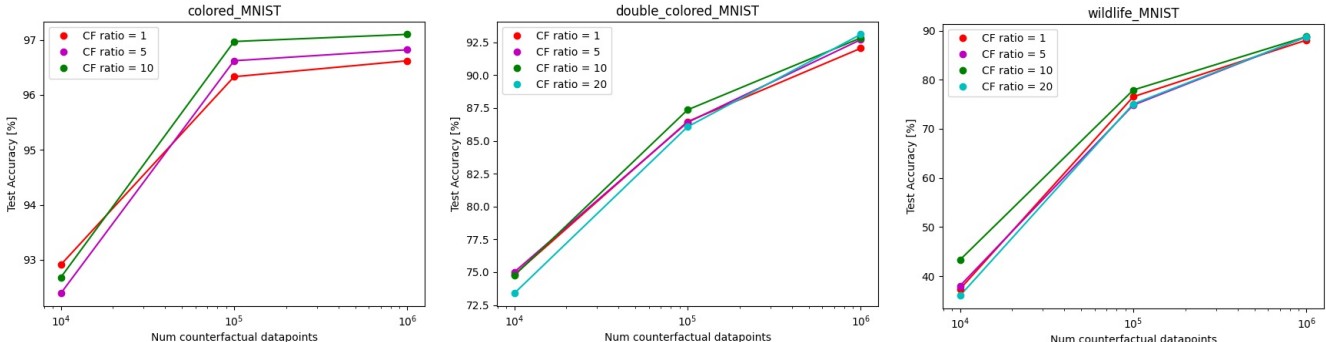

Figure 6: Test accuracy for classifiers trained on original data and counterfactual data with different CF ratios. The CF ratio indicates how many counterfactuals we generate per sampled noise. For colored MNIST, the maximum CF ratio is ten as there are only ten possible colors per shape. The counterfactuals in the trainingdata were generated by a CGN with weights provided by the authors.

 # D   ImageNet hyperparameter search

| Trained on | Lr | CF-ratio | Top-1 Test Accuracies | | | |
| | | | Epoch | IN-9 | Mixed-Same | Mixed-Rand | BG-Gap |
|---|---|---|---|---|---|---|---|
| ImageNet (base) | | 0.0 | 0 | 94.7 % | 85.6 % | 78.1 % | 7.5 % |
| IN-mini | 1E-4 | 0.0 | 16 | 91.6 % | 81.8% | 73.3 % | 8.5 % |
| | 1E-3 | 0.0 | 22 | 90.7 % | 79.6 % | 69.6 % | 10.0 % |
| IN-mini + CGN | 1E-4 | 1.0 | 11 | 90.9 % | 81.8 % | 73.5 % | 8.3 % |
| | 1E-4 | 2.0 | 20 | 91.6 % | 81.9 % | 74.2 % | 7.7 % |
| | 1E-3 | 1.0 | 18 | 88.7 % | 79.9 % | 71.4 % | 8.5 % |
| | 1E-3 | 2.0 | 28 | 88.7 % | 78.9 % | 70.6 % | 8.3 % |
| IN-mini + our CGN | 1E-4 | 1.0 | 17 | 89.7 % | 81.3 % | 72.7 % | 8.6 % |
| | 1E-4 | 2.0 | 16 | 91.4 % | 81.7 % | 72.3 % | 9.4 % |

Table 7: **ImageNet-9 classification hyperparameters**. Investigated hyperparameters for the pre-trained ResNet-50 model.

# E    ImageNet CGN

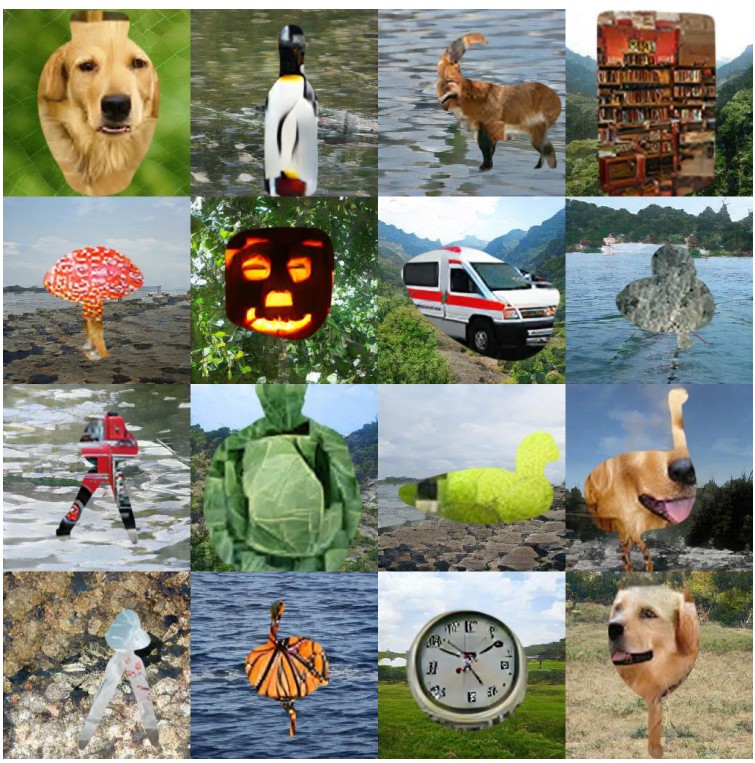

| Row | Column | Shape | Texture | Background |
|---|---|---|---|---|
| 1 | 1 | vase | golden retriever | black and gold garden spider |
| 1 | 2 | wine bottle | king penguin | beaver |
| 1 | 3 | hartebeest | red wolf | drake |
| 1 | 4 | wardrobe | bookshop | valley |
| 2 | 1 | mushroom | agaric | breakwater |
| 2 | 2 | toaster | jack-o'-lantern | jackfruit |
| 2 | 3 | tub | ambulance | valley |
| 2 | 4 | drake | megalith | paddle |

| Row | Column | Shape | Texture | Background |
|---|---|---|---|---|
| 3 | 1 | tripod | fire engine | beaver |
| 3 | 2 | sweatshirt | head cabbage | cliff |
| 3 | 3 | drake | tennis ball | breakwater |
| 3 | 4 | ostrich | golden retriever | geyser |
| 4 | 1 | tripod | plastic bag | rock crab |
| 4 | 2 | ostrich | monarch butterfly | grey whale |
| 4 | 3 | orange | analog clock | viaduct |
| 4 | 4 | red wine | golden retriever | ostrich |

Figure 7: **Counterfactual images.** Here, counterfactual images generated by the CGN, that we trained, are displayed. In the table below the image, labels given to each independent mechanism can be found.

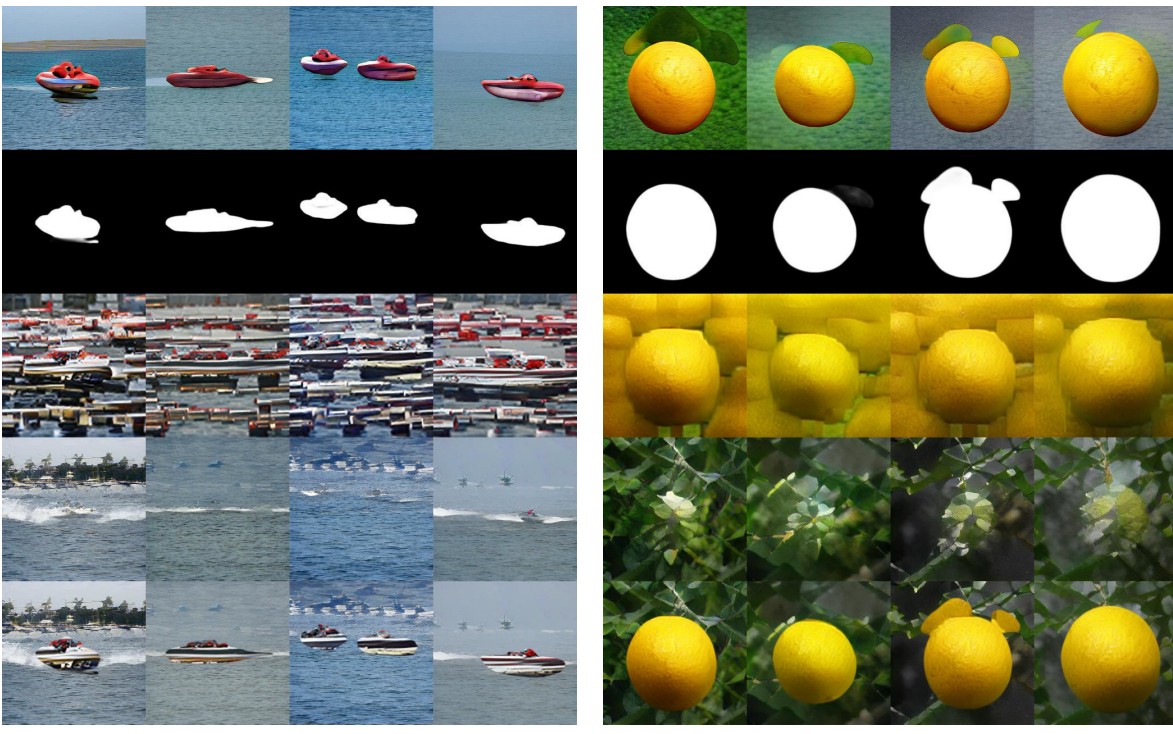

(a) **IM outputs for 'boat'.** From top to bottom: $\tilde{\mathbf{m}}$, $\mathbf{m}$, $\mathbf{f}$, $\mathbf{b}$, $\mathbf{x_{gen}}$

(b) **IM outputs for 'lemon.'** From top to bottom: $\tilde{\mathbf{m}}$, $\mathbf{m}$, $\mathbf{f}$, $\mathbf{b}$, $\mathbf{x_{gen}}$

Figure 8: **IM outputs:** Output of the each mechanism in Fig 1 is displayed. From top to bottom; pre-trained BigGAN, $f_{shape}$, $f_{texture}$, $f_{background}$ output and finally the composition mechanism's output ($x_{gen}$) is displayed.

# F Mechanism and failures of ImageNet CGN

In this section, we discuss the three supposedly independent components of the CGN and their failure cases.

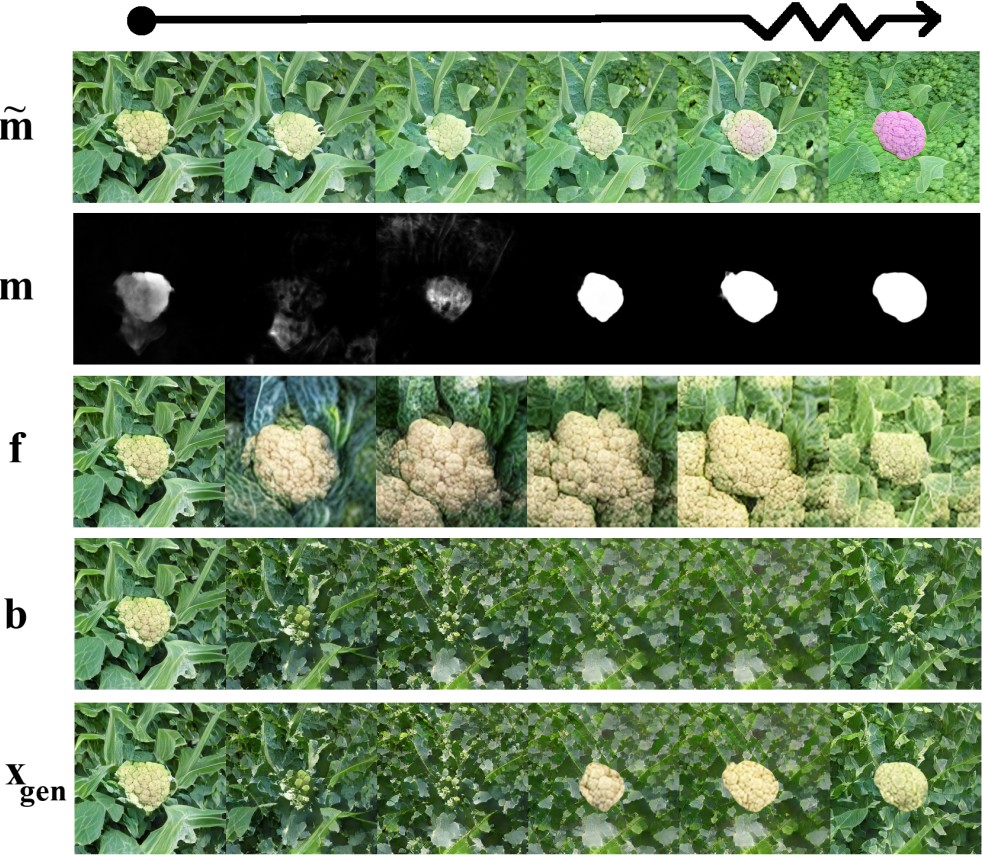

Figure 9: **IM Outputs for cauliflower** Learnt pre-masks $\tilde{\mathbf{m}}$, masks $\mathbf{m}$, foregrounds $\mathbf{f}$, and backgrounds $\mathbf{b}$. The arrow indicates the beginning of training till the jumped ending.

The figure above indicates visually that the individual losses cause intended changes in the BigGAN for each mechanism. Pre-mask $\tilde{\mathbf{m}}$ accentuates the location of the object, foreground $\mathbf{f}$ shows near complete texture mapping, some background artifacts remain in the texture which occures often across classes. Mask $b$ always converges well with no remaining object across the classes, although object artifacts do occur when training the network longer.

Failure cases of the CGN are displayed under three main categories; texture-background entanglement, background residue and reduced realism. All those images are generated using the CGN we trained.

We conjecture that most texture-background entanglement occurs when the mask of the object is either difficult to learn, is relatively small $\mu_{mask} < 0.1$ with $\tau = 0.1$, or the patch size does not match the object well. All of these cases cause the object mask to include more background for lower loss, resulting in texture-background entanglement.

Background residue occurs when background loss has converged during training. The background BigGAN then further improves by decreasing the reconstruction loss. In increasing the background details, that are outside the mask, the inside of the mask is allowed to change in all ways as long as no object is detected. This results in undetected artifacts that are increasingly present in longer trained models.

With a perfect object mask, texture and background identical realism to the BigGAN is expected. The visual results and Inception Score confirm that this is not the case. Reduced realism is prevalent amongst all generated images and are conjectured to be caused by several issues. Which are an entangled problem between the mechanisms.

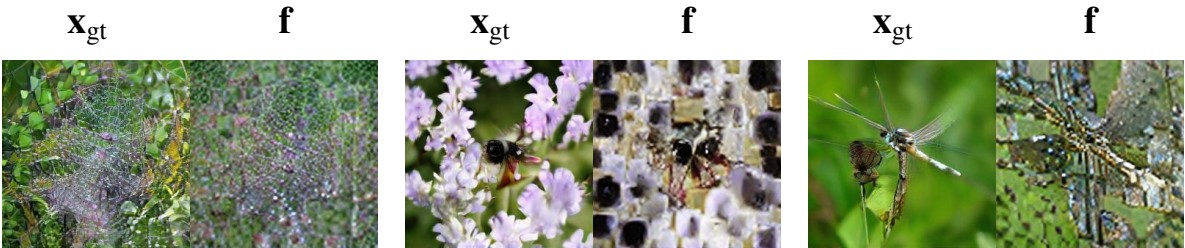

Figure 10: **Texture-Background Entanglement**. Texture maps contain traces from the background. From left to right, the images are; spiderweb, bee and dragonfly.

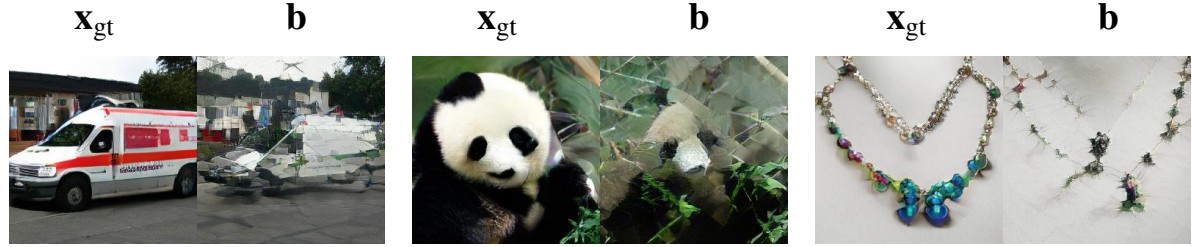

Figure 11: **Background Residue**. Regions, where the objects are located, are not fully in-painted. Background still contains some artifacts from the object.

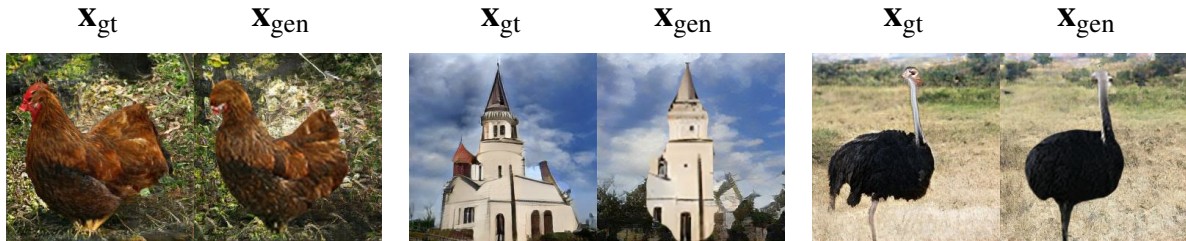

Figure 12: **Reduced realism.** Generated images mostly do not look realistic. Authors stated that this is due to the constraints enforced and the analytically defined composition mechanism.

