# OpenReview forum: "Reproducibility study - Counterfactual Generative Networks"
_ML_Reproducibility_Challenge/2021/Fall — Reject_

### Official Review · Reviewer_kJhj · 2022-02-28
**Interesting work, more details could be provided on hyper parameters and ablation**

**Rating:** 7
**Confidence:** 5

**Review:**

With respect to the formatting, the first page is present and the template stylesheet was used for the paper.
The paper satisfactorily states the scope of reproducibility, and adheres to it.
With respect to whether reproduced from scratch or re-used author repository: The hyperparameter search is shown in Table 7 . Additional codebase with readable code/docs was provided but was derived from the one the original authors code

The authors did communicate with the original authors.
The hyperparameter search is shown in Table 7 but there is not a description of what was done and reported in Table 7, at least not to a degree that provides insights of the search.
The ablation study is provided in Figure 5, but the only description is the one in the figure caption.
The report provides a good detailed discussion of the results and the reproducibility .
The report also details the difficult and easy aspects to reproduce from the original paper.
The authors of the report do not directly address the original authors but their recommendations can be used by the original authors to improve the reproducibility of the paper.
Finally, the reproducibility paper is well organized and easy to follow. Just a few typos need correction there are missing numbers in:  page 4 Section ?? and page 8 Table ??

---

### Official Review · Reviewer_2Ew2 · 2022-03-07
**The report's clarity need to be improved.**

**Rating:** 5
**Confidence:** 3

**Review:**

- The scope of reproducibility is clearly presented.

- The author re-uses the original paper's repository. The hyperparameters are also directly borrowed from the original paper. The author does not perform an additional hyper-parameter search.

- The submission provides a discussion on the verification of the three claims from the original paper.

- Due to limited resources, the report experiments with ImageNet-mini instead of the ImageNet used in the original paper.

- The paper's clarity can be improved. (1) there are multiple places that have "??", e.g., line 100, line 238. (2) missing result in Table 4.

---

### Meta-Review · Program_Chairs · 2022-04-09

**Recommendation:** Reject
**Confidence:** 5

**Metareview:**

Per the reviews, the paper could benefit from additional hyperparameter search.  Additional details regarding hyperparameter search and ablation would also be helpful to the paper. The paper is not accepted.

---

### Decision · Program_Chairs · 2022-04-09

Reject